# Perpetration of and Victimization in Cyberbullying and Traditional Bullying in Adolescents with Attention-Deficit/Hyperactivity Disorder: Roles of Impulsivity, Frustration Intolerance, and Hostility

**DOI:** 10.3390/ijerph18136872

**Published:** 2021-06-26

**Authors:** Tai-Ling Liu, Ray C. Hsiao, Wen-Jiun Chou, Cheng-Fang Yen

**Affiliations:** 1Department of Psychiatry, School of Medicine, College of Medicine, Kaohsiung Medical University, Kaohsiung 80708, Taiwan; tlliu@kmu.edu.tw; 2Department of Psychiatry, Kaohsiung Medical University Hospital, Kaohsiung 80708, Taiwan; 3Department of Psychiatry and Behavioral Sciences, University of Washington School of Medicine, Seattle, WA 98195-6560, USA; rhsiao@u.washington.edu; 4Department of Psychiatry, Children’s Hospital and Regional Medical Center, Seattle, WA 98105, USA; 5School of Medicine, Chang Gung University, Taoyuan 33302, Taiwan; 6Department of Child and Adolescent Psychiatry, Chang Gung Memorial Hospital, Kaohsiung Medical Center, Kaohsiung 83301, Taiwan

**Keywords:** adolescent, attention-deficit hyperactivity disorder, cyberbullying, frustration discomfort, hostility, impulsivity, traditional bullying

## Abstract

Victimization and perpetration of cyberbullying and traditional bullying are prevalent among adolescents with attention-deficit/hyperactivity disorder (ADHD). This study examined the associations of impulsivity, frustration discomfort, and hostility with victimization and with the perpetration of cyberbullying and traditional bullying in adolescents with ADHD. Self-reported involvement in cyberbullying and traditional bullying was assessed in 195 adolescents with a clinical diagnosis of ADHD. Adolescents also completed questionnaires for impulsivity, frustration discomfort, and hostility. Caregivers completed the Child Behavior Checklist for adolescents’ ADHD, internalization, oppositional defiance, and problems with conduct. The associations of impulsivity, frustration discomfort, and hostility with victimization and perpetration of cyberbullying and traditional bullying were examined using logistic regression analysis. The results demonstrated that after the effects of demographic characteristics and behavioral problems were controlled for, frustration intolerance increased the risks of being cyberbullying victims and perpetrators whereas hostility increased the risks of being the victims and perpetrators of traditional bullying. Impulsivity was not significantly associated with any type of bullying involvement. Prevention and intervention programs should alleviate frustration intolerance and hostility among adolescents with ADHD.

## 1. Introduction

The experiences of victimization and perpetration of cyberbullying and traditional bullying are prevalent among children and adolescents with attention-deficit/hyperactivity disorder (ADHD). Cyberbullying refers to the bullying behaviors perpetrated through electronic means [1]. Traditional bullying can involve physical acts, verbal utterances, social exclusion, property theft, and other behaviors [2]. Compared with those without ADHD, adolescents with ADHD have higher risks of being victims and perpetrators of cyberbullying [3,4,5] and traditional bullying [6,7,8]. Victimization in cyberbullying is significantly associated with depression and suicidality in adolescents with ADHD [4]. Moreover, victimization in and perpetration of traditional bullying are significantly associated with depression in adolescents with ADHD [9]. Therefore, the factors predicting the involvement in cyberbullying and traditional bullying in adolescents with ADHD should be examined, and prevention programs should be developed based on the risk factors identified.

Research has demonstrated that demographic characteristics (i.e., adolescents’ age and parental occupational socioeconomic status), behavioral problems (i.e., symptoms of ADHD and Internet addiction), and behavioral temperament (i.e., low reward responsiveness) predict cyberbullying victimization and perpetration in adolescents with ADHD [4]. Demographic characteristics (i.e., adolescents’ age), behavioral temperament (i.e., high behavioral inhibition and fun seeking), and poor family relationships are the predictors of victimization and perpetration in traditional bullying in adolescents with ADHD [10]. These results of previous studies have supported that the individual and environmental factors relate to adolescents’ involvement in cyberbullying and traditional bullying.

Social-Emotional Development (SED) Model is one of hypothetical models to understand the developmental processes of individuals’ social relationships with others [11]. SED includes understanding, regulating, and expressing emotions in a way that is appropriate for one’s age and development, as well as the ability to establish, maintain, and develop healthy relationships with peers and adults [11]. SED is central to navigating challenges in social interactions in everyday life and to adapting flexibly to situational demands [11]. Research has supported the role of deficits in SED for bullying involvement in adolescents [12,13,14,15]. The present study examined the roles of three social-emotional difficulties, including impulsivity, frustration discomfort, and hostility for victimization in and perpetration of both cyberbullying and traditional bullying in adolescents with ADHD. All impulsivity [16], frustration discomfort [17,18,19], and hostility [20,21] are psychological and behavioral characteristics of difficulties in emotional regulation and expression commonly found in adolescents with ADHD. However, their relationships with involvement in cyberbullying and traditional bullying have not been examined in adolescents with ADHD.

Impulsivity is characterized by the underestimation of harm, nonreflective responses, difficult-to-control desires, and repetitive behaviors to obtain pleasure and gratification [22]. Research has demonstrated that high impulsivity is significantly associated with self-harm [23], craving for illicit substances [24], and Internet addiction [25]. High impulsivity has also been linked to cyberbullying [26] and bullying perpetration [27,28,29,30,31]. Impulsivity is one symptom of ADHD [32]. Whether impulsivity also predicts the involvement in cyberbullying and traditional bullying in adolescents receiving treatment for ADHD warrants further study.

Frustration intolerance refers to the difficulty in accepting that reality does not correspond to one’s personal desires [33]. Research has demonstrated that frustration intolerance is significantly related to depression [34], anxiety [35], substance dependence [36], and Internet addiction [37]. Because frustration intolerance is a type of irrational belief related to emotional and behavioral problems [38] and because it is associated with problems of self-control [39], it is reasonable to hypothesize that adolescents with ADHD with high frustration intolerance have a high risk of perpetrating cyberbullying or traditional bullying. However, the relationships of frustration intolerance with victimization in and perpetration of both cyberbullying and traditional bullying in adolescents with ADHD have not been examined.

Hostility is an emotional state that indicates the intention to harm an individual, and it also denotes expressive characteristics that indicate the potential intent for physical aggression and assault [40]. Hostility is associated with mental health problems such as depression [41], suicide [42], and Internet addiction [43]. Research has found that hostility mediates the relationship between prior bullying victimization and subsequent bullying perpetration [44]. Hostility also increases the risk of the co-occurrence of traditional bullying and cyberbullying [45]. However, the role of hostility in victimization in cyberbullying and traditional bullying in adolescents with ADHD has not been examined.

This study examined the associations of impulsivity, frustration discomfort, and hostility with victimization in and perpetration of both cyberbullying and traditional bullying in adolescents with ADHD. As illustrated in Figure 1, we hypothesized that high impulsivity, frustration discomfort, and hostility are significantly associated with the risks of being victims and perpetrators of cyberbullying and traditional bullying in adolescents with ADHD.

## 2. Methods

### 2.1. Participants

Adolescents aged between 11 and 18 years who visited the child and adolescent psychiatric outpatient clinics of two medical centers and who had been diagnosed with ADHD according to the DSM-5 [32] were consecutively invited to participate in this study from June 2019 to January 2021. ADHD was diagnosed based on the outcomes of diagnostic interviews with adolescents and caregivers that were conducted by child psychiatrists. Adolescents and caregivers with intellectual disabilities, schizophrenia, bipolar disorder, autistic disorder, communication difficulties, or cognitive deficits that adversely affected their ability to understand the study’s purpose or complete the questionnaires were excluded. A total of 208 adolescents who had been diagnosed with ADHD and their caregivers were selected for this study; 195 (93.8%) agreed to participate in this study. This study was approved by the Institutional Review Boards of Kaohsiung Medical University (KMUHIRB-SV(I)-20190034) and Chang Gung Memorial Hospital, Kaohsiung Medical Center (201900432A3). Written informed consent was obtained from all participants before assessment.

### 2.2. Measures

#### 2.2.1. Cyberbullying Experience Questionnaire

Adolescents reported their experiences of cyberbullying perpetration and victimization on social media (Facebook, Twitter, and Plurk) or through pictures or video clips, emails, or blogs in the previous year using the six-item cyberbullying experience questionnaire [4]. Each item is scored on a 4-point Likert scale ranging from 0 (*never*) to 3 (*all the time*). The first three items for cyberbullying perpetration are related to the posting of mean or hurtful comments, the posting of upsetting pictures, photos, or videos, and the spreading of rumors online. The final three items are related to the experiences of cyberbullying victimization resulting from the actions described in the first three items. The Cronbach’s α values of the items for cyberbullying victimization and perpetration were 0.70 and 0.65, respectively. Participants with a score of 1, 2, or 3 on any item among the first and final three items were defined to be as self-reported perpetrators and victims of cyberbullying, respectively.

#### 2.2.2. Chinese Version of the School Bullying Experience Questionnaire

In this study, adolescents reported their experiences of traditional bullying victimization and perpetration in the previous year using the Chinese version of the 16-item School Bullying Experience Questionnaire (C-SBEQ) [46,47]. Each item is scored on a 4-point Likert scale ranging from 0 (*never*) to 3 (*all the time*). The first eight items evaluate the experiences of victimization from social, verbal, and physical bullying; the final eight items evaluate the experiences of perpetrating bullying in terms of actions that perpetrate social, verbal, and physical bullying (in the first eight items). Participants who scored 2 or 3 on any item among the first and final eight items were defined to be victims and perpetrators of traditional bullying, respectively. A previous study demonstrated the reliability and validity of the C-SBEQ [47]. In the present study, the Cronbach’s α values of the items for victimization and perpetration of traditional bullying were 0.76 and 0.72, respectively, for adolescents with ADHD.

#### 2.2.3. Impulsivity

In this study, the Barratt Impulsiveness Scale version 11-Taiwan (BIS-11-TW) was used to evaluate the self-reported level of adolescents’ impulsivity [22,48]. The BIS-11-TW contains 25 items measuring multiple aspects of impulsivity, including the inability to plan, the lack of foresight and perseverance and self-control, and a proclivity toward seeking novelty and hastily making decisions. All items are measured on a 4-point scale ranging from 1 (*rarely/never*) to 4 (*almost always/always*), with a higher score indicating higher impulsivity. The BIS-11-TW has acceptable psychometric properties [22]. The Cronbach’s α of the BIS-11-TW was 0.90 in the present study.

#### 2.2.4. Frustration Intolerance

The Chinese version of the Frustration Discomfort Scale (FDS) was used to evaluate the self-reported frustration intolerance of adolescents [39,49,50]. The FDS contains 28 items evaluated on a 5-point Likert scale, with scores ranging from 28 to 140; a higher total score indicates higher frustration intolerance. The Cronbach’s α of the FDS was 0.88 in the present study.

#### 2.2.5. Hostility

The Buss–Durkee Hostility Inventory-Chinese version-Short Form (BDHIC-SF) was used to measure the self-reported level of adolescents’ hostility [51,52]. The BDHIC-SF comprises 20 items measuring hostility cognition, hostility affect, expressive hostility behavior, and suppressive hostility behavior. These items are rated from 1 (*strongly disagree*) to 5 (*strongly agree*), with a higher total score indicating higher hostility. The BDHIC-SF has acceptable psychometric properties [52]. In this study, the Cronbach’s α of the BDHIC-SF was 0.85.

#### 2.2.6. Child Behavior Checklist for Ages 6–18

The 112-item caregiver-reported Chinese version of the Child Behavior Checklist for Ages 6–18 (CBCL/6-18) was used to measure adolescents’ behavior problems [53,54,55]. We used the recommended *T*-score transformations of raw behavior scores, which were adjusted for age and sex differences in behavior found in normative samples. We used the domains of internalizing problems (which includes scales for anxiety/depression, withdrawal/depression, and somatic complaint syndrome), ADHD, oppositional defiant disorder (ODD), and conduct symptoms for analysis.

#### 2.2.7. Demographic Characteristics

The present study examined adolescents’ age and sex and caregivers’ age, sex, educational duration, marital status (married and living together vs. divorced or separated), and occupational socioeconomic status. We assessed the caregivers’ occupational socioeconomic status using the Close-Ended Questionnaire of the Occupational Survey [56], which classifies paternal and maternal occupational socioeconomic status into five levels (level 1 to level 5). A higher level indicates a higher occupational socioeconomic status.

### 2.3. Statistical Analysis

The descriptive statistics used were the frequency and percentage for categorical variables and as the mean and standard deviation (SD) for continuous variables. Correlations between the variables were examined using Spearman’s rank-order correlation. Because of multiple comparisons, the significance level for Spearman’s rank-order correlation was adjusted with *p* < 0.003 (0.05/13) indicating the significance.

Demographic characteristics, impulsivity, frustration intolerance, hostility, and behavioral problems were compared between victims and nonvictims and between perpetrators and nonperpetrators using the *t*-test and chi-square test. The variables with significant differences were included in a logistic regression with conditional forward selection to examine their associations with involvements in cyberbullying and traditional bullying. Odds ratios and 95% confidence intervals were used to represent statistical significance. A two-tailed *p* value of <0.05 indicated statistical significance.

## 3. Results

Table 1 presents the data on demographic characteristics, cyberbullying and bullying involvement, impulsivity, frustration intolerance, hostility, and behavioral problems. In total, 31 girls and 164 boys participated in this study. Their mean age was 13.5 years (SD = 2.3 years). Among them, 14.4% were victims of cyberbullying, 8.7% were perpetrators of cyberbullying, 27.7% were victims of traditional bullying, and 17.9% were perpetrators of traditional bullying.

Table 2 presents the results of Spearman’s rank-order correlation examining the correlations between studied variables. The significant correlations included: negative correlation between child’s age and ODD problems; positive correlations of being cyberbullying victims with being cyberbullying perpetrators and frustration intolerance; positive correlation between being cyberbullying perpetrators and ODD symptoms; positive correlations of being traditional bullying victims with being traditional bullying perpetrators, hostility, and internalizing problems; positive correlations of being traditional bullying perpetrators with frustration intolerance and hostility; positive correlations of impulsivity with frustration intolerance, hostility, ADHD, ODD and conduct problems; positive correlation between frustration intolerance and hostility; positive correlation between hostility and ODD symptoms; positive correlation of internalizing problems with ADHD, ODD and conduct problems; positive correlation of ADHD problems with ODD and conduct problems; and positive correlation between ODD and conduct problems.

Table 3 presents the differences in demographic characteristics, impulsivity, frustration intolerance, hostility, and behavioral problems between victims and nonvictims and between perpetrators and nonperpetrators. The results demonstrated that cyberbullying victims had higher frustration intolerance and hostility relative to cyberbullying nonvictims. Cyberbullying perpetrators had higher ODD problems and frustration intolerance relative to cyberbullying nonperpetrators. Traditional bullying victims were younger and had higher internalizing problems, impulsivity, and hostility relative to traditional bullying nonvictims. Traditional bullying perpetrators had higher impulsivity, frustration intolerance, and hostility relative to traditional bullying nonperpetrators.

The significant variables were included in logistic regression analysis with conditional forward selection as independent variables to examine their associations with involvements in cyberbullying and traditional bullying (Table 4). The results demonstrated that after demographic characteristics and behavioral problems were controlled for, frustration intolerance increased the risks of being cyberbullying victims and perpetrators, whereas hostility increased the risks of being traditional bullying victims and perpetrators. Impulsivity was not significantly associated with any type of bullying involvement.

## 4. Discussion

The present study shows that impulsivity, frustration intolerance, and hostility play various roles in the bullying involvement of adolescents with ADHD. Frustration intolerance increased the risks of being cyberbullying victims and perpetrators, whereas hostility increased the risks of being traditional bullying victims and perpetrators. Impulsivity was not significantly associated with any type of bullying involvement.

Adolescents with ADHD with higher frustration intolerance had higher risks of being cyberbullying victims and perpetrators. Adolescents with ADHD with high frustration intolerance may have difficulties in making and maintaining peer relationships [57,58] and in achieving adequate academic performance [59]. Research has shown that frustration intolerance is associated with behavior avoidance [39]. Adolescents with ADHD may feel uncomfortable and spend more time and energy on Internet activities to feel a sense of achievement [60], increasing the risk of Internet addiction [37]. Internet addiction is a well-known risk factor for cyberbullying perpetration [4] and victimization [61]. Moreover, adolescents with ADHD with high frustration intolerance may work off their frustration by perpetrating cyberbullying; their acts of cyberbullying may provoke others to fight back and thus increase the perpetrator’s risk of victimization. The results of the present study indicate that interventions are required in adolescents with ADHD with high frustration intolerance to reduce their risk of involvement in cyberbullying. Research has found that overlapping cognitive deficits may underlie both classical ADHD symptoms, such as low frustration tolerance and mood instability; stimulants or atomoxetine may alleviate both types of symptoms when they co-occur [62,63]. Supportive interventions [61], rational emotive behavior therapy [64], and cognitive behavioral therapies [65] may also alleviate ADHD youths’ emotional instability due to frustration intolerance.

Consonant with previous studies [44,45], the present study demonstrates that adolescents with ADHD with high hostility had higher risks of being victims and perpetrators of traditional bullying. Individuals with high hostility may enact their intention to harm others in the form of physical and verbal aggression. Moreover, victims of traditional bullying may feel unsafe, increasing their awareness of possible harassment from others at any moment; thus, their hostility may increase. Adolescents with ADHD with high hostility may also overreact to the ordinary words and behaviors of others; thus; they have an increased risk of victimization. Although the result of this study demonstrated the necessity of implementing interventions for hostility in adolescents with ADHD, alleviating hostility in this patient group is a challenge. Previous studies have found no evidence for the efficacies of methylphenidate [66,67] or atomoxetine [68]. Furthermore, no formal, evidence-based model of psychotherapy is available for reducing hostility in individuals with high hostility.

Because high hostility has been found to increase the risk of Internet addiction [43] and because Internet addiction increases the risk of cyberbullying involvement in adolescents with ADHD [37], it is reasonable to hypothesize that hostility may increase the risks of victimization in and perpetration of cyberbullying in adolescents with ADHD. However, the results of this study did not support this hypothesis. Research examining the difference between hostility in the real world and online has demonstrated that the level of hostility is lower online than offline [69]. This may partially account for the different roles of hostility in the involvement in cyberbullying and traditional bullying.

Impulsivity is associated with reactive aggression in impulsive adolescents who cannot control emotion and delay gratification [70]. It is reasonable to hypothesize that ADHD adolescents with high impulsivity may perpetrate bullying due to dissatisfaction when interacting with others. Moreover, ADHD adolescents with high impulsivity may have difficulties in following the rules and thus become the target of being rejected by others. However, incongruent with our hypothesis and the results of previous studies [26,27,28,29,30,31], our findings did not demonstrate a significant association between impulsivity and any type of bullying involvement. The first possible reason accounting for the discrepancy between the result and the hypothesis is the source of participants recruited into this study. Adolescents in this study were recruited from clinical units and are currently receiving medication or psychotherapy for their ADHD. Research has supported the effectiveness of methylphenidate and atomoxetine on reducing the core symptoms of ADHD [71,72,73]. Thus, impulsivity and related problems might be attenuated. Second, victims and perpetrators of traditional bullying tended to have higher impulsivity, respectively; however, the difference in impulsivity became nonsignificant in the multivariate logistic regression analysis. This result indicates that in adolescents receiving treatment for ADHD, other factors such as hostility may play a more significant role in traditional bullying involvement than impulsivity does. Impulsivity indicates a difficult-to-control desire to obtain pleasure and gratification [16]. Contrarily, hostility indicates the intention to harm others and denotes expressive intent for physical aggression and assault [34]. According to Dan Olweus, bullying is a negative action occurred when a person intentionally inflicts injury or discomfort upon another person [74]. Therefore, hostility may play a more direct and stronger role in perpetrating bullying compared with impulsivity.

To our knowledge, this study is one of the first studies to examine the relationships of impulsivity, frustration intolerance, and hostility with involvement in cyberbullying and traditional bullying among adolescents with ADHD. However, several limitations of this study warrant being addressed. First, the causal relationships of impulsivity, frustration intolerance, and hostility with bullying involvement could not be determined in this cross-sectional study. Second, the data of this study were provided by the adolescents except for behavioral problems reported by caregivers, which might result in shared-method variance. Third, the participants of this study were recruited from outpatient clinics; the results of this study might not be generalized to adolescents who did not visit medical units for help. Fourth, research revealed that stimulants such as methylphenidate can lessen involvement in the bullying cycle in adolescents with ADHD [75], whereas the effect of non-stimulants such as atomoxetine on reducing the risk of bullying involvement is still clear. We did not collect the kinds of pharmacological and psychological treatments that the adolescents with ADHD received and could not determine the treatment effects on the associations between impulsivity, frustration discomfort, and hostility with victimization in and perpetration of bullying in adolescents with ADHD.

## 5. Conclusions

The present study shows that frustration intolerance and hostility are significantly associated with various types of bullying involvement of adolescents with ADHD. Mental health professionals should take frustration intolerance and hostility into consideration when developing prevention and intervention strategies to reduce the risks of victimization and perpetration in cyberbullying and traditional bullying among adolescents with ADHD. Mental health professionals should evaluate routinely the levels of frustration intolerance and hostility in adolescents with ADHD. The possibility of involvement in bullying victimization and perpetration should be monitored among those with high levels of frustration intolerance and hostility. Necessary psychological and pharmacological intervention should be provided for adolescents with ADHD to increase frustration tolerance and reduce hostility and the risks of bullying involvement.

## Figures and Tables

**Figure 1 ijerph-18-06872-f001:**
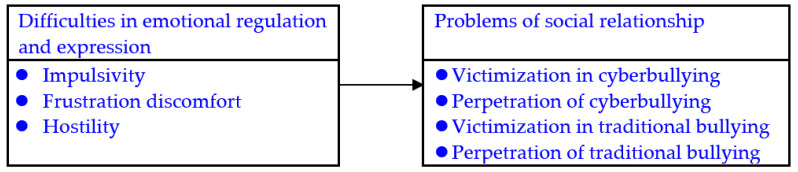
Hypothetical frame of this study.

**Table 1 ijerph-18-06872-t001:** Demographic characteristics, bullying involvement, impulsivity, frustration intolerance, hostility, and behavioral problems (N = 195).

Variable	*n* (%)	Mean (SD)	Range
Adolescents			
Sex			
Female	31 (15.9)		
Male	164 (84.1)		
Age (years)		13.5 (2.3)	11–18
Cyberbullying			
Victims	28 (14.4)		
Perpetrators	17 (8.7)		
Bullying			
Victims	54 (27.7)		
Perpetrators	35 (17.9)		
Impulsivity on the BIS-11		61.5 (9.7)	35–94
Frustration discomfort on the FDS		68.4 (24.1)	28–129
Hostility on the BDSI-CS		55.2 (16.4)	20–94
Child Behavior Checklist			
Internalizing problems		58.7 (10.6)	33–83
Attention-deficit/hyperactivity problems		61.7 (7.6)	40–80
Oppositional defiant problems		59.5 (7.9)	50–80
Conduct problems		57.9 (8.3)	50–86
Caregivers			
Sex			
Female	156 (80)		
Male	39 (20)		
Age (years)		45.4 (6.4)	31–69
Educational level (years)		14.1 (3.2)	6–28
Marriage status			
Married and living together	155 (79.5)		
Divorced or separated	40 (20.5)		
Paternal occupational socioeconomic status		2.7 (1.0)	1–5
Maternal occupational socioeconomic status		2.2 (1.1)	1–5

BDHIC-SF: Buss-Durkee Hostility Inventory-Chinese version-Short Form; BIS-11: Barratt Impulsiveness Scale version 11; FDS: Frustration Discomfort Scale; SD: standard deviation.

**Table 2 ijerph-18-06872-t002:** Correlation between studied variables: Spearman’s rank-order correlation.

Variables	Spearman’s Rho Coefficient												
	1.	2.	3.	4.	5.	6.	7.	8.	9.	10.	11.	12.	13.
1. Adolescents’ sex	1												
2. Adolescents’ age	−0.110	1											
3. Cyberbullying victims	0.058	0.089	1										
4. Cyberbullying perpetrators	0.085	0.011	**0.496**	1									
5. Bullying victims	0.050	−0.177	0.139	0.093	1								
6. Bullying perpetrators	0.094	−0.084	0.151	0.045	**0.457**	1							
7. Impulsivity	0.069	−0.181	0.135	0.134	0.192	0.205	1						
8. Frustration intolerance	0.023	0.057	**0.237**	0.201	0.144	**0.216**	**0.347**	1					
9. Hostility	−0.077	−0.038	0.203	0.183	**0.242**	**0.276**	**0.492**	**0.682**	1				
10. Internalizing problems	−0.059	−0.053	0.008	0.038	**0.227**	0.124	0.170	0.097	0.158	1			
11. ADHD problems	−0.050	−0.145	0.023	0.108	0.128	0.108	**0.252**	0.033	0.085	**0.533**	1		
12. ODD problems	0.128	**−0.239**	0.055	**0.209**	0.083	0.060	**0.243**	0.156	**0.259**	**0.518**	**0.591**	1	
13. Conduct problems	−0.042	−0.144	0.009	0.103	0.110	0.102	**0.209**	0.126	0.180	**0.605**	**0.648**	**0.697**	1

ADHD: attention-deficit/hyperactivity disorder; ODD: oppositional defiant disorder. Values in bold are significant at the *p* < 0.003 level.

**Table 3 ijerph-18-06872-t003:** Differences in demographic characteristics, impulsivity, frustration intolerance, hostility, and behavioral problems between victims and nonvictims and between perpetrators and nonperpetrators.

Variable	Cyberbullying Victims				Cyberbullying Perpetrators				Bullying Victims				Bullying Perpetrators			
	Yes	No	χ^2^ or *t*	*p*	Yes	No	χ^2^ or *t*	*p*	Yes	No	χ^2^ or *t*	*p*	Yes	No	χ^2^ or *t*	*p*
Sex ^a^																
Female	3 (9.7)	28 (90.3)	0.657	0.418	1 (3.2)	30 (96.8)	1.397	0.237	3 (9.7)	28 (90.3)	1.712	0.191	7 (22.6)	24 (77.4)	0.481	0.188
Male	25 (15.2)	139 (84.8)			16 (9.8)	148 (90.2)			32 (19.5)	132 (80.5)			47 (28.7)	117 (71.3)		
Age (years) ^b^	13.9 (2.2)	13.4 (2.3)	−1.029	0.305	13.6 (2.3)	13.5 (2.3)	−0.133	0.895	12.9 (2.0)	13.8 (2.3)	2.558	0.011	13.1 (2.1)	13.6 (2.3)	1.239	0.217
Child Behavior Checklist ^b^																
Internalizing problems	58.4 (10.3)	58.8 (10.7)	0.177	0.860	59.5 (10.6)	58.7 (10.7)	−0.302	0.763	62.5 (10.2)	57.3 (10.5)	−3.118	0.002	61.5 (11.1)	58.1 (10.5)	−1.721	0.087
ADHD problems	62.4 (8.3)	61.6 (7.5)	−0.480	0.632	64.8 (7.7)	61.4 (7.5)	−1.740	0.084	63.1 (8.2)	61.2 (7.3)	−1.634	0.104	63.8 (8.1)	61.3 (7.4)	−1.800	0.073
ODD problems	60.6 (8.1)	59.3 (7.9)	−0.819	0.414	65.8 (9.3)	58.9 (7.5)	−3.543	<0.001	60.8 (8.7)	59.0 (7.6)	−1.451	0.148	60.3 (7.7)	59.3 (8.0)	−0.641	0.522
Conduct problems	58.6 (9.4)	57.8 (8.2)	−0.455	0.650	61.6 (10.9)	57.6 (8.0)	−1.934	0.055	59.4 (9.1)	57.4 (8.0)	−1.525	0.129	59.9 (9.1)	57.5 (8.1)	−1.506	0.134
Impulsivity ^b^	64.1 (5.7)	61.0 (10.2)	−1.540	0.125	65.2 (6.1)	61.1 (9.9)	−1.656	0.099	64.5 (8.3)	60.3 (10.0)	−2.781	0.006	65.3 (7.5)	60.6 (10.0)	−2.627	0.009
Frustration intolerance ^b^	81.5 (18.7)	66.2 (24.2)	−3.175	0.002	82.7 (16.5)	67.0 (24.3)	−2.602	0.010	73.4 (24.1)	66.5 (23.9)	−1.809	0.072	78.4 (18.0)	66.2 (24.7)	−2.759	0.006
Hostility^b^	63.0 (12.3)	53.8 (16.7)	−2.777	0.006	64.1 (11.1)	54.3 (16.6)	−2.366	0.019	61.4 (16.4)	52.8 (15.8)	−3.376	0.001	64.9 (15.2)	53.0 (16.0)	−4.042	<0.001

ADHD: attention-deficit/hyperactivity disorder; ODD: oppositional defiant disorder; SD: standard deviation. ^a^: *n* (%); ^b^: mean (SD).

**Table 4 ijerph-18-06872-t004:** Factors related to being cyberbullying and traditional bullying perpetrators and victims: Logistic regression analysis with conditional forward selection; CI: confidence interval; ODD: oppositional defiant disorder; OR: odds ratio.

Variable	Cyberbullying Victims			Cyberbullying Perpetrators			Bullying Victims			Bullying Perpetrators		
	Wals χ^2^	*p*	OR (95% CI)	Wals χ^2^	*p*	OR (95% CI)	Wals χ^2^	*p*	OR (95% CI)	Wals χ^2^	*p*	OR (95% CI)
Age							5.914	0.015	0.821 (0.700–0.962)			
Internalizing problems							6.928	0.008	1.048 (1.012–1.085)			
ODD problems				9.185	0.002	1.102 (1.035–1.174)						
Frustration intolerance	9.031	0.003	1.028 (1.010–1.047)	5.080	0.024	1.027 (1.003–1.051)						
Hostility							8.375	0.004	1.032 (1.010–1.054)	13.854	<0.001	1.049 (1.023–1.076)

## Data Availability

The data will be available upon reasonable request to the corresponding authors.

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
