# Peer review of "Perpetration of and Victimization in Cyberbullying and Traditional Bullying in Adolescents with Attention-Deficit/Hyperactivity Disorder: Roles of Impulsivity, Frustration Intolerance, and Hostility"

_ijerph, 2021, doi:10.3390/ijerph18136872_

Round 1
Reviewer 1 Report
I suggest the authors:
(i) improving the structure of the introduction. First of all eliminating the subsection number (1.1, 1.2, and 1.3) and adjusting the contents in order to have a unique section. I also suggest shifting the contents of section 1.3 at the beginning of the introduction.
(ii) reducing the number of sub-sections of section 2.2 and adjusting the contents in order to have a maximum of 2-3 subsections)
(iii) improving the Results section better explaining the correlation of the analysed variables
iv) reducing the number of Discussion section and adjusting the contents in order to have a unique section
Author Response
Comment 1
I suggest the authors improving the structure of the introduction. First of all eliminating the subsection number (1.1, 1.2, and 1.3) and adjusting the contents in order to have a unique section. I also suggest shifting the contents of section 1.3 at the beginning of the introduction.
Response
Thank you for your suggestion. In the revised manuscript we eliminated the subsection number (1.1, 1.2, and 1.3) and moved the contents of subsection 1.3 forward. We also added the theoretical model (Social-emotional development) as well as a figure illustrating the hypothetical frame of this study (please refer to Figure 1). Please refer to line 57-73.
“…These results of previous studies have supported that the individual and environmental factors relate to adolescents’ involvement in cyberbullying and traditional bullying.
Social-Emotional Development (SED) Model is one of hypothetical models to understand the developmental processes of individuals’ social relationships with others [11]. SED includes understanding, regulating, and expressing emotions in a way that is appropriate for one’s age and development, as well as the ability to establish, maintain, and develop healthy relationships with peers and adults [11]. SED is central to navigating challenges in social interactions in everyday life and to adapting flexibly to situational demands [11]. Research has supported the role of deficits in SED for bullying involvement in adolescents [12-15]. The present study examined the roles of three social-emotional difficulties, including impulsivity, frustration discomfort, and hostility for victimization in and perpetration of both cyberbullying and traditional bullying in adolescents with ADHD. All impulsivity [16], frustration discomfort [17-19], and hostility [20,21] are psychological and behavioral characteristics of difficulties in emotional regulation and expression commonly found in adolescents with ADHD. However, their relationships with involvement in cyberbullying and traditional bullying have not been examined in adolescents with ADHD.”
Comment 2
(ii) reducing the number of sub-sections of section 2.2 and adjusting the contents in order to have a maximum of 2-3 subsections)
Response
We reducing the number of sub-sections of subsection 2.2. Please refer to line 125-188.
Comment 3
(iii) improving the Results section better explaining the correlation of the analysed variables
Response
Thank you for your suggestion. We added the result of correlation tests between the studied variables into the revised manuscript as below.
Methods
“Correlations between the variables were examined using Spearman's rank-order correlation. Because of multiple comparisons, the significance level for Spearman's rank-order correlation was adjusted with p < 0.003 (0.05/13) indicating the significance.” Please refer to line 191-194.
Results
“Table 2 presents the results of Spearman's rank-order correlation examining the correlations between studied variables. The significant correlations included: negative correlation between child’s age and ODD problem; positive correlations of being cyberbullying victims with being cyberbullying perpetrators and frustration intolerance; positive correlation between being cyberbullying perpetrators and ODD symptoms; positive correlations of being traditional bullying victims with being traditional bullying perpetrators, hostility, and internalizing problems; positive correlations of being traditional bullying perpetrators with frustration intolerance and hostility; positive correlations of impulsivity with frustration intolerance, hostility, ADHD, ODD and conduct problems; positive correlation between frustration intolerance and hostility; positive correlation between hostility and ODD symptoms; positive correlation of internalizing problems with ADHD, ODD and conduct problems; positive correlation of ADHD problems with ODD and conduct problems; and positive correlation between ODD and conduct problems.” Please refer to line 213-225.
Comment 4
- iv) reducing the number of Discussion section and adjusting the contents in order to have a unique section
Response
Thank you for your suggestion. We reduced the number of Discussion section. Please refer to line 252-335. We also added more discussion into the revised manuscript as below. Please refer to line 297-319.
“Impulsivity is associated with reactive aggression in impulsive adolescents who cannot control emotion and delay gratification [70]. It is reasonable to hypothesize that ADHD adolescents with high impulsivity may perpetrate bullying due to dissatisfaction when interacting with others. Moreover, ADHD adolescents with high impulsivity may have the difficulties in following the rules and thus become the target of be rejected by others. However, incongruent with our hypothesis and the results of previous studies [26-31], our findings did not demonstrate a significant association between impulsivity and any type of bullying involvement. The first possible reason accounting for the discrepancy between the result and the hypothesis is the source of participants recruited into this study. Adolescents in this study were recruited from clinical units and are currently receiving medication or psychotherapy for their ADHD. Research has supported the effectiveness of methylphenidate and atomoxetine on reducing the core symptoms of ADHD [71-73]. Thus, impulsivity and related problems might be attenuated. Second, victims and perpetrators of traditional bullying tended to have higher impulsivity, respectively; however, the difference in impulsivity became nonsignificant in the multivariate logistic regression analysis. This result indicates that in adolescents receiving treatment for ADHD, other factors such as hostility may play a more significant role in traditional bullying involvement than impulsivity does. Impulsivity indicates a difficult-to-control desire obtain pleasure and gratification [16]. Contrarily, hostility indicates the intention to harm others and denotes expressive intent for physical aggression and assault [34]. According to Dan Olweus, bullying is a negative action occurred when a person intentionally inflicts injury or discomfort upon another person [74]. Therefore, hostility may play a more direct and stronger drive for perpetrating bullying compared with impulsivity.”
Reviewer 2 Report
The paper points out a target of our times the cyberbullism. It is really interesting the correlation with ADHD pathology and the conclusions about the predictor factors for victimazation. I suggest that te conclusion about hostility and traditional bullying could be wider discussed. The conclusion too could be improved, it is very short.
Author Response
Comment
I suggest that the conclusion about hostility and traditional bullying could be wider discussed. The conclusion too could be improved, it is very short.
Response
Thank you for your suggestion. In the revised manuscript we added more discussion about impulsivity, hostility, and bullying into Discussion section as below. Moreover, we rewrote and improved the content of Conclusion section as below.
“…Impulsivity indicates a difficult-to-control desire obtain pleasure and gratification [16]. Contrarily, hostility indicates the intention to harm others and denotes expressive intent for physical aggression and assault [34]. According to Dan Olweus, bullying is a negative action occurred when a person intentionally inflicts injury or discomfort upon another person [74]. Therefore, hostility may play a more direct and stronger drive for perpetrating bullying compared with impulsivity.” Please refer to line 314-319.
“Fourth, research revealed that stimulants such as methylphenidate can lessen involvement in the bullying cycle in adolescents with ADHD [75], whereas the effect of non-stimulants such as atomoxetine on reducing the risk of bullying involvement is still clear. We did not collect the kinds of pharmacological and psychological treatments that the adolescents with ADHD received and could not determine the treatment effects on the associations between impulsivity, frustration discomfort, and hostility with victimization in and perpetration of bullying in adolescents with ADHD.” Please refer to line 329-335.
“Mental health professionals should evaluate routinely the levels of frustration intolerance and hostility in adolescents with ADHD. The possibility of involvement in bullying victimization and perpetration should be monitored among those with high levels of frustration intolerance and hostility. Necessary psychological and pharmacological intervention should be provided for adolescents with ADHD to increase frustration tolerance and reduce hostility and the risks of bullying involvement.” Please refer to line 342-347.
Reviewer 3 Report
An excellent paper, original and novel in its conceptualization, scientifically well-informed and of interest to the reader.
It is fascinating to read how frustration intolerance contributes to cyberbullying, a major issue for today's adolescents, as well as hostility to traditional bullying.
The unexpected finding that impulsivity does not relate to any type of bullying could be more elaborated in comparison to frustration intolerance and hostility, since it is defined as"reactive aggression to gratification delay" which sounds close to frustration intolerance, as well as to the concept of hostility where adolescents " tend to overreact to behaviors of others". The argument that impulsivity may be attenuated by ADHD medical treatment and psychotherapy does not suffice to explain the simultaneous resilience of hostility and frustration intolerance as symptoms. Which was the medication the adolescents were receiving at the time of the cross-sectional study? Could it have different effects on the three studied dimensions? I would like some more elaboration on the interchange of the three concepts in the discussion section.
Author Response
Comment
The unexpected finding that impulsivity does not relate to any type of bullying could be more elaborated in comparison to frustration intolerance and hostility, since it is defined as "reactive aggression to gratification delay" which sounds close to frustration intolerance, as well as to the concept of hostility where adolescents "tend to overreact to behaviors of others". The argument that impulsivity may be attenuated by ADHD medical treatment and psychotherapy does not suffice to explain the simultaneous resilience of hostility and frustration intolerance as symptoms. Which was the medication the adolescents were receiving at the time of the cross-sectional study? Could it have different effects on the three studied dimensions? I would like some more elaboration on the interchange of the three concepts in the discussion section.
Response
Thank you for your comment.
- We added more discussion about the role of impulsivity for bullying involvement as below into the revised manuscript. Especially, we discussed the possible difference in the roles of impulsivity and hostility for bullying involvement. Please refer to line 297-319.
- We agreed that it is important to examine whether various medications may have different effects on impulsivity; however, we did not collect the kinds of medication the participants received. We listed it as one of limitations in this study as below. Please refer to line 329-335.
- “Impulsivity is associated with reactive aggression in impulsive adolescents who cannot control emotion and delay gratification [70]. It is reasonable to hypothesize that ADHD adolescents with high impulsivity may perpetrate bullying due to dissatisfaction when interacting with others. Moreover, ADHD adolescents with high impulsivity may have the difficulties in following the rules and thus become the target of be rejected by others. However, incongruent with our hypothesis and the results of previous studies [26-31], our findings did not demonstrate a significant association between impulsivity and any type of bullying involvement. The first possible reason accounting for the discrepancy between the result and the hypothesis is the source of participants recruited into this study. Adolescents in this study were recruited from clinical units and are currently receiving medication or psychotherapy for their ADHD. Research has supported the effectiveness of methylphenidate and atomoxetine on reducing the core symptoms of ADHD [71-73]. Thus, impulsivity and related problems might be attenuated. Second, victims and perpetrators of traditional bullying tended to have higher impulsivity, respectively; however, the difference in impulsivity became nonsignificant in the multivariate logistic regression analysis. This result indicates that in adolescents receiving treatment for ADHD, other factors such as hostility may play a more significant role in traditional bullying involvement than impulsivity does. Impulsivity indicates a difficult-to-control desire obtain pleasure and gratification [16]. Contrarily, hostility indicates the intention to harm others and denotes expressive intent for physical aggression and assault [34]. According to Dan Olweus, bullying is a negative action occurred when a person intentionally inflicts injury or discomfort upon another person [74]. Therefore, hostility may play a more direct and stronger drive for perpetrating bullying compared with impulsivity.”
“Fourth, research revealed that stimulants such as methylphenidate can lessen involvement in the bullying cycle in adolescents with ADHD [75], whereas the effect of non-stimulants such as atomoxetine on reducing the risk of bullying involvement is still clear. We did not collect the kinds of pharmacological and psychological treatments that the adolescents with ADHD received and could not determine the treatment effects on the associations between impulsivity, frustration discomfort, and hostility with victimization in and perpetration of bullying in adolescents with ADHD.”
Reviewer 4 Report
This is an interesting essay, which will make a very noteworthy contribution to the field. It was annotated and informed by a diverse range of other cognate studies (but I will discuss its later).
Background. In general, the study quite well introduces the issue raised. Nevertheless, I think that the contribution of this article to literature would be considerably greater if the authors presented the adopted theoretical model, explained it using a figure/graph which would also demonstrate the assumed mutual relationships between the examined
characteristics, and subsequently tested this model. While explaining the model, the authors could elucidate their specific hypotheses.
Method. I also have some comments about some other problems - the total lack of any descriptive statistics. Have the basic socio-demographic data been collected concerning the examined childrem and their families? These characteristics seem to be quite important, e.g. education of the parents, economic status of the family, etc.
References. There is a need to focus on recent studies. In you references list are only 6 (!) studies, younger than 5 years.
Author Response
Comment 1
Background. In general, the study quite well introduces the issue raised. Nevertheless, I think that the contribution of this article to literature would be considerably greater if the authors presented the adopted theoretical model, explained it using a figure/graph which would also demonstrate the assumed mutual relationships between the examined characteristics, and subsequently tested this model. While explaining the model, the authors could elucidate their specific hypotheses.
Response
Thank you for your suggestion. We revised the contents of Introduction section by adding the adopted theoretical model (Social-emotional development) as well as a figure illustrating the hypothetical frame of this study as below. We also added the theoretical model (Social-emotional development) as well as a figure illustrating the hypothetical frame of this study (please refer to Figure 1). Please refer to line 57-73.
“…These results of previous studies have supported that the individual and environmental factors relate to adolescents’ involvement in cyberbullying and traditional bullying.
Social-Emotional Development (SED) Model is one of hypothetical models to understand the developmental processes of individuals’ social relationships with others [11]. SED includes understanding, regulating, and expressing emotions in a way that is appropriate for one’s age and development, as well as the ability to establish, maintain, and develop healthy relationships with peers and adults [11]. SED is central to navigating challenges in social interactions in everyday life and to adapting flexibly to situational demands [11]. Research has supported the role of deficits in SED for bullying involvement in adolescents [12-15]. The present study examined the roles of three social-emotional difficulties, including impulsivity, frustration discomfort, and hostility for victimization in and perpetration of both cyberbullying and traditional bullying in adolescents with ADHD. All impulsivity [16], frustration discomfort [17-19], and hostility [20,21] are psychological and behavioral characteristics of difficulties in emotional regulation and expression commonly found in adolescents with ADHD. However, their relationships with involvement in cyberbullying and traditional bullying have not been examined in adolescents with ADHD.”
Comment 2
Method. I also have some comments about some other problems - the total lack of any descriptive statistics. Have the basic socio-demographic data been collected concerning the examined children and their families? These characteristics seem to be quite important, e.g. education of the parents, economic status of the family, etc.
Response
Thank you for your suggestion. In addition to children’s sex and age that have been described in Table 1, we added the data caregivers’ sex, age, educational duration, marital status, and occupational socioeconomic status into the revised manuscript as below. Please refer to Please refer to line 182-188 and Table 1.
“Demographic Characteristics
The present study examined adolescents’ age and sex and caregivers’ age, sex, educational duration, marital status (married and living together vs. divorced or separated), and occupational socioeconomic status. We assessed the caregivers’ occupational socioeconomic status using the Close-Ended Questionnaire of the Occupational Survey [56], which classifies paternal and maternal occupational socioeconomic status into five levels (level 1 to level 5). A higher level indicates a higher occupational socioeconomic status.”
Comment 3
References. There is a need to focus on recent studies. In you references list are only 6 (!) studies, younger than 5 years.
Response
Thank you for your suggestion. We added 7 new citations published in recent 3 years as below into the revised manuscript.
- Agley, J.; Jun, M.; Eldridge, L.; Agley, D.L.; Xiao, Y.; Sussman, S.; Golzarri-Arroyo, L.; Dickinson, S.L.; Jayawardene, W.; Gassman, R. Effects of ACT Out! Social issue theater on social-emotional competence and bullying in youth and adolescents: Cluster randomized controlled trial. JMIR Ment Health. 2021, 8, e25860. doi: 10.2196/25860.
- Dawson, A.E.; Wymbs, B.T.; Evans, S.W.; DuPaul, G.J. Exploring how adolescents with ADHD use and interact with technology. J Adolesc. 2019, 71, 119-137. doi: 10.1016/j.adolescence.2019.01.004.
- Liu, T.L.; Hsiao, R.C.; Chou, W.J.; Yen, C.F. Social anxiety in victimization and perpetration of cyberbullying and traditional bullying in adolescents with autism spectrum disorder and attention-deficit/hyperactivity disorder. Int J Environ Res Public Health. 2021, 18, 5728. doi: 10.3390/ijerph18115728.
- Nickerson, A.B.; Fredrick, S.S.; Allen, K.P.; Jenkins, L.N. Social emotional learning (SEL) practices in schools: Effects on perceptions of bullying victimization. J Sch Psychol. 2019, 73, 74-88. doi: 10.1016/j.jsp.2019.03.002.
- Olweus, D.; Solberg, M.E.; Breivik, K. Long-term school-level effects of the Olweus Bullying Prevention Program (OBPP). Scand J Psychol. 2020, 61, 108-116. doi: 10.1111/sjop.12486.
- Tural Hesapcıoglu, S.; Kandemir, G. Association of methylphenidate use and traditional and cyberbullying in adolescents with ADHD. Pediatr Int. 2020, 62, 725-735. doi: 10.1111/ped.14185. PMID: 32022957.
- Yang, C.; Chan, M.K.; Ma, T.L. School-wide social emotional learning (SEL) and bullying victimization: Moderating role of school climate in elementary, middle, and high schools. J Sch Psychol. 2020, 82, 49-69. doi: 10.1016/j.jsp.2020.08.002.
Round 2
Reviewer 1 Report
The required revisions have been made